# Attribute-Enhanced Similarity Ranking for Sparse Link Prediction

## Abstract

Link prediction is a fundamental problem in graph data. In its most realistic setting, the problem consists of predicting missing or future links between random pairs of nodes from the set of disconnected pairs. Graph Neural Networks (GNNs) have become the predominant framework for link prediction. GNN-based methods treat link prediction as a binary classification problem and handle the extreme class imbalance—real graphs are very sparse—by sampling (uniformly at random) a balanced number of disconnected pairs not only for training but also for evaluation. However, we show that the reported performance of GNNs for link prediction in the balanced setting does not translate to the more realistic imbalanced setting and that simpler topology-based approaches are often better at handling sparsity. These findings motivate Gelato, a similarity-based link-prediction method that applies (1) graph learning based on node attributes to enhance a topological heuristic, (2) a ranking loss for addressing class imbalance, and (3) a negative sampling scheme that efficiently selects hard training pairs via graph partitioning. Experiments show that Gelato is more accurate and faster than GNN-based alternatives.

## 1    Introduction

Machine learning on graphs supports various structured-data applications including social network analysis (Tang et al., 2008; Li et al., 2017; Qiu et al., 2018a), recommender systems (Jamali and Ester, 2009; Monti et al., 2017; Wang et al., 2019a), natural language processing (Sun et al., 2018a; Sahu et al., 2019; Yao et al., 2019), and physics modeling (Sanchez-Gonzalez et al., 2018; Ivanovic and Pavone, 2019; da Silva et al., 2020). Among the graph-related tasks, one could argue that link prediction, which consists of predicting missing or future links (Lü and Zhou, 2011; Martínez et al., 2016), is the most fundamental one. This is because link prediction not only has many concrete applications (Qi et al., 2006; Liben-Nowell and Kleinberg, 2007) but can also be considered an (implicit or explicit) step of the graph-based machine learning pipeline (Martin et al., 2016; Bahulkar et al., 2018; Wilder et al., 2019)—as the observed graph is usually noisy and/or incomplete.

Graph Neural Networks (GNNs) (Kipf and Welling, 2017; Hamilton et al., 2017; Veličković et al., 2018) have emerged as the predominant paradigm for machine learning on graphs. Similar to their great success in node classification (Klicpera et al., 2018; Wu et al., 2019; Zheng et al., 2020) and graph classification (Ying et al., 2018; Zhang et al., 2018a; Morris et al., 2019), GNNs have been shown to achieve state-of-the-art link prediction performance (Zhang and Chen, 2018; Liu et al., 2020; Pan et al., 2022; Yun et al., 2021; Chamberlain et al., 2023; Wang et al., 2023). Compared to classical approaches that rely on expert-designed heuristics to extract topological information (e.g., Common Neighbors (Newman, 2001), Adamic-Adar (Adamic and Adar, 2003), Preferential Attachment (Barabási et al., 2002)), GNNs can naturally incorporate attributes and are believed to be able to learn new effective heuristics directly from data via supervised learning.

However, we argue that *the evaluation of GNN-based link prediction methods paints an overly optimistic view of their model performance*. Most real graphs are sparse and have a modular structure (Barabási, 2016; Newman, 2018). In Cora and Citeseer (citation networks), less than 0.2% of the node pairs are links/positive (see Table 1) and modules arise around research topics. Yet GNN-based link prediction methods are evaluated on an artificially balanced test set that includes every positive pair but only a small sample of the negative ones chosen uniformly at random (Hu et al., 2020). Due to modularity, the majority of negative pairs sampled are expected to be relatively far from each other

(i.e. across different modules) compared to positive pairs. As a consequence, performance metrics reported for this balanced setting, which we call *biased testing*, differ widely from the ones observed for the more challenging *unbiased testing*, where the test set includes every disconnected pair of nodes. In particular, we have found that unsupervised topological heuristics are more competitive in the *unbiased setting*, often outperforming recent GNN-based link prediction methods. This finding has motivated us to rethink the design of link prediction methods for sparse graphs.

A key hypothesis of our work is that effective unbiased link prediction in sparse graphs requires a similarity metric that is able to distinguish positive pairs from hard negative ones. More specifically, link prediction should be seen as a "needle in the haystack" type of problem, where extreme class imbalance makes even the most similar pairs still more likely to be negative. Existing GNN-based approaches fail in this sparse regime due to (1) their use of a binary classification loss that is highly sensitive to class imbalance; (2) their *biased training* that mimics *biased testing*; (3) their inability to learn effective topological heuristics directly from data.

The goal of this paper is to address the key limitations of GNNs for link prediction mentioned above. We present *Gelato*, a novel similarity-based framework for link prediction that combines a topological heuristic and graph learning to leverage both topological and attribute information. Gelato applies a ranking-based N-pair loss and partitioning-based negative sampling to select hard training node pairs. Extensive experiments demonstrate that our model significantly outperforms state-of-the-art GNN-based methods in both accuracy and scalability. Figure 1 provides an overview of our approach.

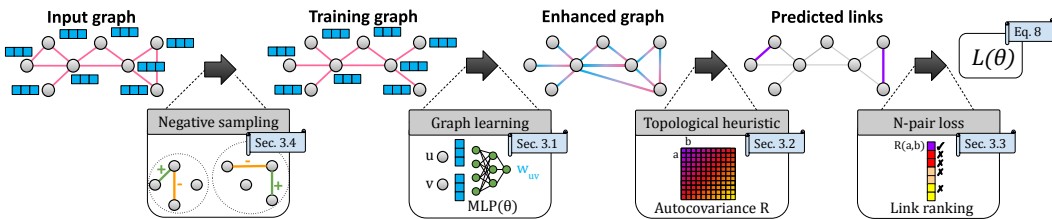

Figure 1: Gelato applies graph learning to incorporate attribute information into the topology. The learned graph is given to a topological heuristic that predicts edges between node pairs with high Autocovariance similarity. The parameters of the MLP are optimized end-to-end using the N-pair loss over node pairs selected via a partitioning-based negative sampling scheme. Experiments show that Gelato outperforms state-of-the-art GNN-based link prediction methods.

To summarize, our contributions are: (1) We scrutinize the evaluation of supervised link prediction methods and identify their limitations in handling class imbalance; (2) we propose a simple, effective, and efficient framework to combine topological and attribute information for link prediction; (3) we introduce an N-pair link prediction loss that we show to be more effective at addressing class imbalance; and (4) we propose an efficient partitioning-based negative sampling scheme that improves link prediction generalization in the sparse setting.

## 2 LIMITATIONS IN SUPERVISED LINK PREDICTION EVALUATION

Supervised link prediction is often formulated as a binary classification problem, where the positive (or negative) class includes node pairs connected (or not connected) by a link. A key difference between link prediction and other classification problems is that the two classes in link prediction are *extremely* imbalanced since most graphs of interest are sparse (see Table 1). However, the class imbalance is not properly addressed in the evaluation of existing supervised approaches.

Existing link prediction methods (Kipf and Welling, 2016; Zhang and Chen, 2018; Chami et al., 2019; Zhang et al., 2021; Cai et al., 2021; Yan et al., 2021; Zhu et al., 2021; Chen et al., 2022; Pan et al., 2022) are evaluated on a test set that contains all positive test pairs and only an equal number of random negative pairs. Similarly, the Open Graph Benchmark (OGB) ranks predicted links against a very small sample of random negative pairs. We term these approaches *biased testing* as they highly overestimate the ratio of positive pairs in the graph. This issue is exacerbated in most real graphs, where community structure (Newman, 2006) causes random negative pairs to be particularly easy to identify—as they likely involve members of different communities. Evaluation metrics based on

these biased test sets provide an overly optimistic measurement of the actual performance in *unbiased testing*, where every negative pair is included in the test set. In fact, in real applications where positive test edges are not known a priori, it is impossible to construct those biased test sets to begin with.

Regarding evaluation metrics, Area Under the Receiver Operating Characteristic Curve (AUC) and Average Precision (AP) are the two most popular evaluation metrics for supervised link prediction (Kipf and Welling, 2016; Zhang and Chen, 2018; Chami et al., 2019; Zhang et al., 2021; Cai et al., 2021; Yan et al., 2021; Zhu et al., 2021; Chen et al., 2022; Pan et al., 2022). We first argue that, as in other imbalanced classification problems (Davis and Goadrich, 2006; Saito and Rehmsmeier, 2015), AUC is not an effective evaluation metric for link prediction as it is biased towards the majority class (non-edges). On the other hand, AP and other rank-based metrics such as Hits@$k$—used in OGB (Hu et al., 2020)—are effective for imbalanced classification *but only if evaluated on an unbiased test*.

*Example*: Consider an instance of Stochastic Block Model (Karrer and Newman, 2011) with 10 blocks of size 1k, intra-block density 0.9, and inter-block density 0.1. The number of inter-block negative pairs is $10 \times 1k \times (10-1) \times 1k \times (1-0.1)/2 = 40.5M$, while the number of intra-block negative pairs, which have high topological similarities like the ground-truth positive pairs and are much harder to contrast against, is $10 \times 1k \times 1k \times (1-0.9)/2 = 0.5M$. Biased testing would select less than $0.5M/(0.5M + 40.5M) < 2\%$ of the test negative pairs among the (hard) intra-block ones.

The above discussion motivates a more representative evaluation setting for supervised link prediction. We argue for the use of rank-based evaluation metrics—AP, Precision@$k$ (Lü and Zhou, 2011), and Hits@$k$ (Bordes et al., 2013)—with *unbiased testing*, where positive edges are ranked against hard negative node pairs. These metrics have been widely applied in related problems, such as unsupervised link prediction (Lü and Zhou, 2011; Ou et al., 2016; Zhang et al., 2018b; Huang et al., 2021), knowledge graph completion (Bordes et al., 2013; Yang et al., 2015; Sun et al., 2018b), and information retrieval (Schütze et al., 2008), where class imbalance is also significant. In our experiments, we will illustrate how these evaluation metrics combined with *unbiased testing* provide a drastically different and more informative performance evaluation compared to existing approaches.

## 3 METHOD

The limitations of supervised link prediction methods, including GNNs, to handle *unbiased testing* in sparse graphs motivate the design of a novel link prediction approach. First, preliminary results (see Table 6) have shown that topological heuristics such as Common Neighbors are not impacted by class imbalance. That is due to the fact that these heuristics are sensitive to small differences in structural similarity between positive and hard negative pairs while not relying on any learning—and thus not being affected by biased training. However, different from GNNs, topological heuristics are unable to leverage node attribute information. Our approach addresses this limitation by integrating supervision into a powerful topological heuristic to leverage attribute data via graph learning.

**Notation and problem.** Consider an attributed graph $G = (V, E, X)$, where $V$ is the set of $n$ nodes, $E$ is the set of $m$ edges (links), and $X = (x_1, ..., x_n)^T \in \mathbb{R}^{n \times r}$ collects $r$-dimensional node attributes. The topological (structural) information of the graph is represented by its adjacency matrix $A \in \mathbb{R}^{n \times n}$, with $A_{uv} > 0$ if an edge of weight $A_{uv}$ connects nodes $u$ and $v$ and $A_{uv} = 0$, otherwise. The (weighted) degree of node $u$ is given as $d_u = \sum_v A_{uv}$ and the corresponding degree vector (matrix) is denoted as $d \in \mathbb{R}^n$ ($D \in \mathbb{R}^{n \times n}$). The volume of the graph is $\text{vol}(G) = \sum_u d_u$. Our goal is to infer missing links in $G$ based on its topological and attribute information, $A$ and $X$.

**Model overview.** Figure 1 provides an overview of our link prediction model. It starts by selecting training node pairs using a novel partitioning-based negative sampling scheme. Next, a topology-centric graph learning phase incorporates node attribute information directly into the graph structure via a Multi-layer Perceptron (MLP). We then apply a topological heuristic, Autocovariance (AC), to the attribute-enhanced graph to obtain a pairwise score matrix. Node pairs with the highest scores are predicted as (positive) links. The scores for training pairs are collected to compute an N-pair loss. Finally, the loss is used to train the MLP parameters in an end-to-end manner. We named our model Gelato (Graph enhancement for link prediction with autocovariance). Gelato represents a different paradigm in supervised link prediction combining a graph encoding of attributes with a topological heuristic instead of relying on node embeddings.

## 3.1 Graph learning

The goal of graph learning is to generate an enhanced graph that incorporates node attribute information into the topology. This can be considered as the "dual" operation of message-passing in GNNs, which incorporates topological information into attributes (embeddings). We argue that graph learning is the more suitable scheme to combine attributes and topology for link prediction since it does not rely on the GNN to learn a topological heuristic, which we have verified empirically to be a challenge.

Specifically, our first step of graph learning is to augment the original edges with a set of node pairs based on their (untrained) attribute similarity (i.e., adding an $\epsilon$-neighborhood graph):

$$\widetilde{E} = E + \{(u, v) \mid s(x_u, x_v) > \epsilon_\eta\} \tag{1}$$

where $s(\cdot)$ can be any similarity function (we use cosine in our experiments) and $\epsilon_\eta$ is a threshold that determines the number of added pairs as a ratio $\eta$ of the original number of edges $m$.

A simple MLP then maps the pairwise node attributes into a trained edge weight for every edge in $\widetilde{E}$:

$$w_{uv} = \text{MLP}([x_u; x_v]; \theta) \tag{2}$$

where $[x_u; x_v]$ denotes the concatenation of $x_u$ and $x_v$ and $\theta$ contains the trainable parameters. For undirected graphs, we instead use the following permutation invariant operator (Chen et al., 2014):

$$w_{uv} = \text{MLP}([x_u + x_v; |x_u - x_v|]; \theta) \tag{3}$$

The final edge weights of the enhanced graph are a weighted combination of the topological weights, the untrained weights, and the trained weights:

$$\widetilde{A}_{uv} = \alpha A_{uv} + (1 - \alpha)(\beta w_{uv} + (1 - \beta)s(x_u, x_v)) \tag{4}$$

where $\alpha$ and $\beta$ are hyperparameters. The enhanced adjacency matrix $\widetilde{A}$ is then fed into a topological heuristic for link prediction introduced in the next section. Note that the MLP is not trained directly to predict the links, but instead trained end-to-end to enhance the input graph given to the topological heuristic. Also note that the MLP can be easily replaced by a more powerful model such as a GNN, but the goal of this paper is to demonstrate the general effectiveness of our framework and we will show that even a simple MLP leads to significant improvement over the base heuristic.

## 3.2 Topological heuristic

Assuming that the learned adjacency matrix $\widetilde{A}$ incorporates structural and attribute information, Gelato applies a topological heuristic to $\widetilde{A}$. Specifically, we adopt Autocovariance, which has been shown to achieve state-of-the-art link prediction results for non-attributed graphs (Huang et al., 2021).

Autocovariance is a random-walk-based similarity metric. Intuitively, it measures the difference between the co-visiting probabilities for a pair of nodes in a truncated walk and in an infinitely long walk. Given the enhanced graph $\widetilde{G}$, the Autocovariance similarity matrix $R \in \mathbb{R}^{n \times n}$ is given as

$$R = \frac{\widetilde{D}}{\text{vol}(\widetilde{G})}(\widetilde{D}^{-1}\widetilde{A})^t - \frac{\tilde{d}\tilde{d}^T}{\text{vol}^2(\widetilde{G})} \tag{5}$$

where $t \in \mathbb{N}_0$ is the scaling parameter of the truncated walk. Each entry $R_{uv}$ represents a similarity score for node pair $(u, v)$ and top similarity pairs are predicted as links. Note that $R_{uv}$ only depends on the $t$-hop enclosing subgraph of $(u, v)$ and can be easily differentiated with respect to the edge weights in the subgraph. In fact, Gelato could be applied with any differentiable topological heuristics or even a combination of them. In our experiments (Section 4.3), we will show that Autocovariance alone enables state-of-the-art link prediction performance without requiring any learning.

**Scaling up Gelato with batching and sparse operations.** Naively implementing Gelato using dense tensors is infeasible, due to the quadratic VRAM requirement ($R \in \mathbb{R}^{|V| \times |V|}$). To address this limitation, we store $\widetilde{A}$ as a sparse matrix. Then, instead of directly computing $(\widetilde{D}^{-1}\widetilde{A})^t$ from Equation 5 (resulting on a dense $|V| \times |V|$ matrix), we compute

$$P_{k+1} = P_k(\widetilde{D}^{-1}\widetilde{A}), \qquad k \in \{1, 2, ..., t\} \tag{6}$$

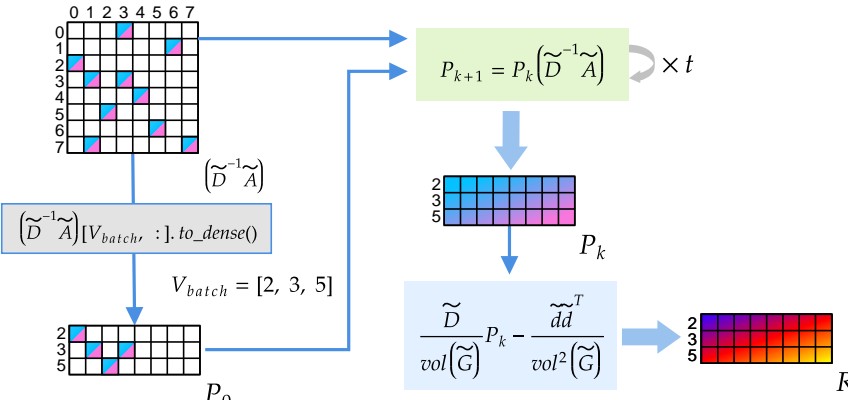

Figure 2: Scaling up Gelato using batching and sparse tensors. Given a batch of nodes ($V_{batch}$), we operate only on a slice ($P_0$) of the enhanced and normalized transition matrix ($\widetilde{D}^{-1}\widetilde{A}$) to compute the Autocovariance matrix slice $R$ for pairs containing nodes in $V_{batch}$. This is implemented efficiently using dense-sparse tensor multiplication, as the transition matrix is expected to be sparse.

$$R = \frac{\widetilde{D}}{\text{vol}(\widetilde{G})} P_t - \frac{\tilde{d}\tilde{d}^T}{\text{vol}^2(\widetilde{G})} \tag{7}$$

where $P_0 = (\widetilde{D}^{-1}\widetilde{A})_{ij}$, for all $i \in V_{batch}$, where $V_{batch}$ consists of the nodes in the current batch. To put it simply, this operation substitution allows us to compute a sequence of $t$ multiplications between a dense $P_k \in \mathbb{R}^{|batch| \times |V|}$ matrix and a sparse matrix $(\widetilde{D}^{-1}\widetilde{A}) \in \mathbb{R}^{|V| \times |V|}$ instead of a dense matrix power operation, $(\widetilde{D}^{-1}\widetilde{A})^t$. The overall VRAM usage is reduced from $O(|V|^2)$ to $O(|batch| \cdot |V|)$.

Next, we introduce how to train our model parameters with supervised information.

### 3.3 N-PAIR LOSS

Supervised link prediction methods rely on the cross entropy loss (CE) to optimize model parameters. However, CE is known to be sensitive to class imbalance (Byrd and Lipton, 2019). Instead, Gelato applies the N-pair loss (Sohn, 2016) that is inspired by the metric learning and learning-to-rank literature (McFee and Lanckriet, 2010; Cakir et al., 2019; Revaud et al., 2019; Wang et al., 2019b) to train the parameters of our graph learning model from highly imbalanced *unbiased training* data.

The N-pair loss (NP) contrasts each positive training edge $(u, v)$ against a set of negative pairs $N(u, v)$. It is computed as follows:

$$L(\theta) = - \sum_{(u,v) \in E} \log \frac{\exp(R_{uv})}{\exp(R_{uv}) + \sum_{(p,q) \in N(u,v)} \exp(R_{pq})} \tag{8}$$

Intuitively, $L(\theta)$ is minimized when each positive edge $(u, v)$ has a much higher similarity than its contrasted negative pairs: $R_{uv} \gg R_{pq}, \forall (p, q) \in N(u, v)$. Compared to CE, NP is more sensitive to negative pairs that have comparable similarities to those of positive pairs—they are more likely to be false positives. While NP achieves good performance in our experiments, alternative losses from the learning-to-rank literature (Freund et al., 2003; Xia et al., 2008; Bruch, 2021) could also be applied.

### 3.4 NEGATIVE SAMPLING

To minimize distribution shifts between training and test, negative samples $N(u, v)$ must be generated using *unbiased training*. This means that $N(u, v)$ is a random subset of all disconnected pairs in the training graph, and $|N(u, v)|$ is proportional to the ratio of negative pairs. In this way, we enforce $N(u, v)$ to include hard negative pairs. However, due to graph sparsity (see Table 1, this

approach does not scale to large graphs as the total number of negative pairs would be $O(|V|^2 - |E|)$. Supervised methods for link prediction bypass this challenge by sampling a small number of negative pairs uniformly at random but most of these pairs are expected to be easy (see Section 2).

To efficiently generate a small number of hard negative pairs, Gelato applies a simple yet novel negative sampling scheme based on graph partitioning (Fortunato, 2010). The idea is to select negative samples inside partitions as they are expected to have similarity values comparable to positive pairs. We apply METIS (Karypis and Kumar, 1998) to obtain $p$ partitions $G_p = (V_p, E_p, X_p), V_p \subset V, E_p \subset E, X_p \subset X$, such that $V = \bigcup_{i=0}^{p} V_i$ and $|V_i| \approx |V|/p$. Then, we apply *unbiased training* only *within each partition*, reducing the number of sampled negative pairs to $|E^-| = \sum_{i}^{p} |V_i|^2 - |E_i|$. In the remainder of the paper, we refer to this approach as *partitioned training*. We claim that this procedure filters (easy) pairs consisting of nodes that would be too far away in the network topology from training while maintaining the more informative (hard) pairs that are closer and topologically similar according to METIS. We include in the Appendix (See Figure 5) a performance comparison between Gelato trained using *unbiased training* against *partitioned training*.

## 4 EXPERIMENTS

In this section, we provide empirical evidence for our claims regarding supervised link prediction and demonstrate the accuracy and efficiency of Gelato. Our implementation is anonymously available at `https://anonymous.4open.science/r/Gelato/`.

### 4.1 EXPERIMENT SETTINGS

**Datasets.** Our method is evaluated on four attributed graphs commonly used for link prediction (Chami et al., 2019; Zhang et al., 2021; Yan et al., 2021; Zhu et al., 2021; Chen et al., 2022; Pan et al., 2022; Hu et al., 2020). Table 1 shows dataset statistics.

Table 1: A summary of dataset statistics.

|  | #Nodes | #Edges | #Attrs | Avg. degree | Density |
|---|---|---|---|---|---|
| CORA | 2,708 | 5,278 | 1,433 | 3.90 | 0.14% |
| CITESEER | 3,327 | 4,552 | 3,703 | 2.74 | 0.08% |
| PUBMED | 19,717 | 44,324 | 500 | 4.50 | 0.02% |
| OGBL-DDI | 4,267 | 1,334,889 | 0 | 500,5 | 7.33% |
| OGBL-COLLAB | 235,868 | 1,285,465 | 128 | 8.2 | 0.0046% |

**Baselines.** For GNN-based link prediction, we include four state-of-the-art methods published in the past two years: Neo-GNN (Yun et al., 2021), BUDDY (Chamberlain et al., 2023), and NCN / NCNC (Wang et al., 2023), as well as the pioneering work—SEAL (Zhang and Chen, 2018). For topological link prediction heuristics, we consider Common Neighbors (CN) (Newman, 2001), Adamic Adar (AA) (Adamic and Adar, 2003), and Autocovariance (AC) (Huang et al., 2021)—the base heuristic in our model.

**Hyperparameters.** For Gelato, we tune the proportion of added edges $\eta$ from $\{0.0, 0.25, 0.5, 0.75, 1.0\}$, the topological weight $\alpha$ from $\{0.0, 0.25, 0.5, 0.75\}$, and the trained weight $\beta$ from $\{0.25, 0.5, 0.75, 1.0\}$. All other settings are fixed across datasets: MLP with one hidden layer of 128 neurons, AC scaling parameter $t = 3$, Adam optimizer (Kingma and Ba, 2015) with a learning rate of 0.001, a dropout rate of 0.5, and *unbiased training* without downsampling. To maintain fairness in our results, we also tuned the baselines and exposed our procedures in detail in the Appendix C. For all models, including Gelato, the tuning process is done in all datasets, with the exception of OGBL-COLLAB.

**Data splits for unbiased training and unbiased testing.** Following Kipf and Welling (2016); Zhang and Chen (2018); Chami et al. (2019); Zhang et al. (2021); Chen et al. (2022); Pan et al. (2022), we adopt 85%/5%/10% ratios for training, validation, and testing. Specifically, for *unbiased training* and *unbiased testing*, we first randomly divide the (positive) edges $E$ of the original graph into $E_{train}^+$, $E_{valid}^+$, and $E_{test}^+$ for training, validation, and testing based on the selected ratios. Then, we set the negative pairs in these three sets as (1) $E_{train}^- = E^- + E_{valid}^+ + E_{test}^+$, (2) $E_{valid}^- = E^- + E_{test}^+$,

and (3) $E_{test}^- = E^-$, where $E^-$ is the set of all negative pairs (excluding self-loops) in the original graph. Notice that the validation and testing *positive* edges are included in the *negative* training set, and the testing *positive* edges are included in the *negative* validation set. This setting simulates the real-world scenario where the test edges (and the validation edges) are unobserved during validation (training). For *negative sampling*, we repeat the dividing procedure above for each generated partition $G_i$. The final sets are unions of individual sets for each partition: $E_{train}^{+/-} = \bigcup_{i=0}^p E_{train_i}^{+/-}$, $E_{valid}^{+/-} = \bigcup_{i=0}^p E_{valid_i}^{+/-}$, and $E_{test}^{+/-} = \bigcup_{i=0}^p E_{test_i}^{+/-}$.

**Evaluation metrics.** We adopt $hits@k$ —the ratio of positive edges individually ranked above $k$th place against all negative pairs—as our evaluation metric since it represents a good notion of class distinction under heavily imbalanced scenarios in information retrieval, compatible with the intuition of link prediction as a similarity-based ranking task.

## 4.2 Partitioned Sampling and Link prediction as a similarity task

This section provides empirical evidence for some of the claims made regarding limitations in the evaluation of supervised link prediction methods (see Section 2). It also demonstrates the effectiveness of Gelato to distinguish true links from hard negative node pairs in sparse graphs.

**Negative sampling for harder pairs.** Based on the hardness of negative pairs, the easiest scenario is the *biased testing*, followed by *unbiased testing* and *partitioned testing*—i.e. with negative pairs inside partitions. This can be verified by Figure 3, which compares the predicted scores of NCN against the similarities computed by Gelato on the test set of CITESEER. *Biased testing*, the easiest and most unrealistic scenario, shows a good separation between positive and negative pairs both in NCN and Gelato. For *unbiased testing*, which is more realistic, Gelato is better at distinguishing positive and negative pairs. Finally, *partitioned testing* present a particular challenge but Gelato still ranks most positive pairs above negative ones. Other GNN-based link prediction approaches have shown similar behaviors to NCN.

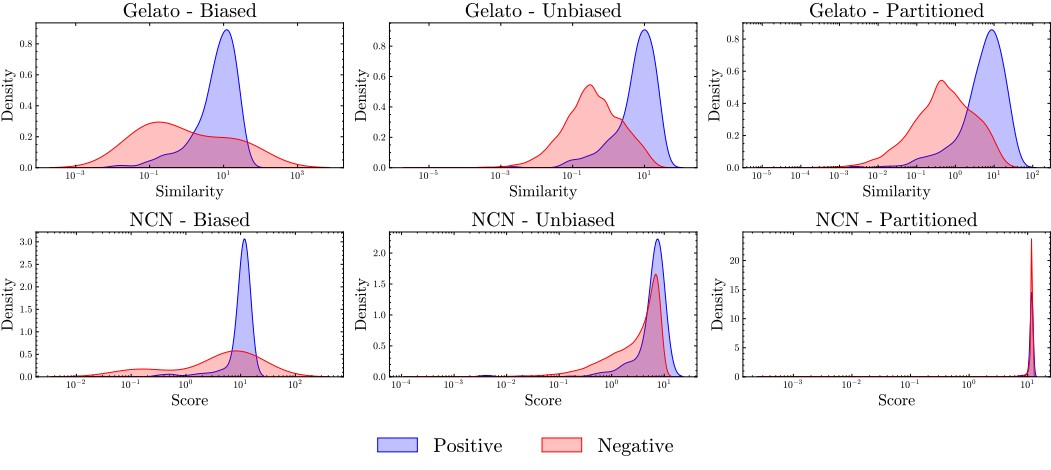

Figure 3: Comparison between the probability density functions of predicted similarities/scores by Gelato and NCN (state-of-the-art GNN), on the test set in three different regimes (biased, unbiased, and partitioned). Negative pairs are represented in red, and positive pairs are represented in blue. Gelato consistently distinguishes between positive and negative pairs across all testing regimes, while NCN struggles as negative pairs become harder.

**Similarity-based link prediction.** Figure 3 shows densities normalized by the size of positive and negative sets, respectively. However, in real-world sparse graphs, the number of negative pairs is much larger than that of positive ones. To better understand the ranking of positive pairs over negative pairs, we also show the same plot with densities normalized by the total number of all pairs in Figure 8 in the Appendix E. The results show that for *unbiased* and *partitioned testing*, ranking positive pairs over hard negative pairs is especially challenging due to their overwhelming number, i.e. positive pairs are "needles in a haystack". This provides evidence that classifiers, such as GNNs for link

prediction, are not suitable for finding decision boundaries in these extremely imbalanced settings, which motivates the design of Gelato as a similarity ranking model trained using an N-pair loss.

### 4.3 LINK PREDICTION PERFORMANCE

Table 2 summarizes the link prediction performance in terms of the mean and standard deviation of $hits@1000$ for all methods. We also include in the Appendix the results of $MRR$ (Mean Reciprocal Rank) and $AP$ (Average Precision) (see Tables 6 and 7).

Table 2: Link prediction performance comparison (mean ± std hits@1000) for all datasets considered. Gelato consistently outperforms GNN-based methods, topological heuristics, and two-stage approaches combining attributes/topology. For CORA, CITESEER, OGBL-DDI and PUBMED results we used *unbiased* training, while for OGBL-COLLAB *negative* sampling is used, for scalability reasons. The top three models are colored by **First**, **Second** and **Third**.

|  |  | CORA | CITESEER | PUBMED | OGBL-DDI | OGBL-COLLAB |
|---|---|---|---|---|---|---|
| GNN | SEAL | 0.0[*] | 7.25[*] | *** | 0.75[*] | 25.9[*] |
|  | Neo-GNN | **6.96 ± 4.24** | 5.42 ± 0.13 | **1.63 ± 0.32** | 0.76[*] | 0.85[*] |
|  | BUDDY | 4.81 ± 0.72 | 5.86 ± 0.34 | OOM | 0.74 ± 0.01 | **27.66 ± 0.24** |
|  | NCN | 4.11 ± 1.22 | 7.84 ± 1.13 | 0.06 ± 0.1 | **0.82 ± 0.02** | 7.16 ± 1.42 |
|  | NCNC | 6.58 ± 0.58 | **8.72 ± 2.08** | 1.04 ± 0.09 | **0.89 ± 0.09** | 0.44 ± 0.37 |
| Topological Heuristics | CN | 4.17 ± 0.00 | 4.4 ± 0.00 | 0.36 ± 0.00 | **0.8 ± 0.00** | 2.4 ± 0.00 |
|  | AA | 6.64 ± 0.00 | 4.4 ± 0.00 | 1.13 ± 0.00 | 0.79 ± 0.00 | 4.88 ± 0.00 |
|  | AC | **11.20 ± 0.00** | **14.29 ± 0.00** | **3.81 ± 0.00** | 0.78 ± 0.00 | 12.89 ± 0.00 |
| Gelato |  | **16.62 ± 0.31** | **19.78 ± 0.23** | **4.18 ± 0.19** | 0.78 ± 0.00 | **30.92**[*] |

[*] Run only once as each run takes >24 hrs;   *** Each run takes >1000 hrs;   OOM: Out Of Memory.

First, we want to highlight the drastically different performance of GNN-based methods compared to those found in the original papers (Zhang and Chen, 2018; Yun et al., 2021; Chamberlain et al., 2023; Wang et al., 2023). Some of them underperform even the simplest topological heuristics such as Common Neighbors under *unbiased testing*. Moreover, Autocovariance, which is the base topological heuristic applied by Gelato and does not account for node attributes, outperforms all the GNN-based baselines for the majority of the datasets. These results support our arguments from Section 2 that evaluation metrics based on *biased testing* can produce misleading results compared to *unbiased testing*. The overall best-performing GNN model is NCNC, which generalizes a pairwise topological heuristic (Common Neighbors) using message-passing. NCNC only outperforms Gelato on OGBL-DDI but Autocovariance achieves even better results. This is an indication that further hyperparameter search will likely improve Gelato's results for this dataset. This is particularly critical in the case of OGBL-DDI, as it is the only dataset considered that does not contain natural node features. Overall, Gelato outperforms the best GNN-based method by **138**%, **125**%, **156**%, and **11**% for CORA, CITESEER, PUBMED, and OGB-COLLAB, respectively. Moreover, Gelato outperforms its base topological heuristic (Autocovariance) by **48**%, **39**%, **10**%, and **139**% for CORA, CITESEER, PUBMED, and OGB-COLLAB, respectively. Additional results are provided in the Appendix F.

## 5 RELATED WORK

**Topological heuristics for link prediction.**  The early link prediction literature focuses on topology-based heuristics. This includes approaches based on local (e.g., Common Neighbors (Newman, 2001), Adamic Adar (Adamic and Adar, 2003), and Resource Allocation (Zhou et al., 2009)) and higher-order (e.g., Katz (Katz, 1953), PageRank (Page et al., 1999), and SimRank (Jeh and Widom, 2002)) information. More recently, random-walk based graph embedding methods, which learn vector representations for nodes (Perozzi et al., 2014; Grover and Leskovec, 2016; Huang et al., 2021), have achieved promising results in graph machine learning tasks. Popular embedding approaches, such as DeepWalk (Perozzi et al., 2014) and node2vec (Grover and Leskovec, 2016), have been shown to implicitly approximate the Pointwise Mutual Information similarity (Qiu et al., 2018b), which can also be used as a link prediction heuristic. This has motivated the investigation

of other similarity metrics such as Autocovariance (Delvenne et al., 2010; Huang et al., 2021; 2022). However, these heuristics are unsupervised and cannot take advantage of data beyond the topology.

**Graph Neural Networks for link prediction.** GNN-based link prediction addresses the limitations of topological heuristics by training a neural network to combine topological and attribute information and potentially learn new heuristics. GAE (Kipf and Welling, 2016) combines a graph convolution network (Kipf and Welling, 2017) and an inner product decoder based on node embeddings for link prediction. SEAL (Zhang and Chen, 2018) models link prediction as a binary subgraph classification problem (edge/non-edge), and follow-up work (e.g., SHHF (Liu et al., 2020), WalkPool (Pan et al., 2022)) investigates different pooling strategies. Other recent approaches for GNN-based link prediction include learning representations in hyperbolic space (e.g., HGCN (Chami et al., 2019), LGCN (Zhang et al., 2021)), generalizing topological heuristics (e.g., Neo-GNN (Yun et al., 2021), NBFNet (Zhu et al., 2021)), and incorporating additional topological features (e.g., TLC-GNN (Yan et al., 2021), BScNets (Chen et al., 2022)). ELPH and BUDDY (Chamberlain et al., 2023) apply hashing to efficiently approximate subgraph-based link prediction models, such as SEAL, using a message-passing neural network (MPNN) with distance-based structural features. More recent, NCNC (Wang et al., 2023) combines the Common Neighbors heuristic with an MPNN achieving state-of-the-art results. Motivated by the growing popularity of GNNs for link prediction, this work investigates key questions regarding their training, evaluation, and ability to learn effective topological heuristics directly from data. We propose Gelato, which is simpler, more accurate, and faster than the state-of-the-art GNN-based link prediction methods.

**Graph learning.** Gelato learns a graph that combines topological and attribute information. Our goal differs from generative models (You et al., 2018; Li et al., 2018; Grover et al., 2019), which learn to sample from a distribution over graphs. Graph learning also enables the application of GNNs when the graph is unavailable, noisy, or incomplete (Zhao et al., 2022). LDS (Franceschi et al., 2019) and GAug (Zhao et al., 2021) jointly learn a probability distribution over edges and GNN parameters. IDGL (Chen et al., 2020) and EGLN (Yang et al., 2021) alternate between optimizing the graph and embeddings for node/graph classification and collaborative filtering. Singh et al. (2021) proposes two-stage link prediction by augmenting the graph as a preprocessing step. In comparison, Gelato effectively learns a graph in an end-to-end manner by minimizing the loss of a topological heuristic.

## 6    CONCLUSION

This work sheds light on key limitations in the evaluation of supervised link prediction methods due to the widespread use of *biased testing*. This has created a consensus within the graph machine learning research community that (1) GNNs are the most promising approach for link prediction, casting topological heuristics obsolete; and (2) link prediction is now an easy problem due to recent advances in deep learning. Our paper challenges both of these assumptions. We show that, when evaluated properly, link prediction in sparse graphs is still a hard problem. In particular, GNNs for link prediction are not effective at handling sparse graphs due to the extreme class imbalance. This has motivated the design of Gelato, a novel link prediction framework introduced in this work.

Gelato is a similarity-based link prediction method that combines graph learning and autocovariance to leverage attribute and topological information. To better handle the class imbalance, Gelato applies an N-pair loss instead of cross-entropy. Finally, to efficiently sample hard negative pairs, we introduce a partitioning-based negative sampling scheme. Extensive experiments show that Gelato is more accurate and scalable than state-of-the-art GNN-based solutions across different datasets.

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

## A    ANALYSIS OF LINK PREDICTION EVALUATION METRICS WITH DIFFERENT TEST SETTINGS

*Example*: Consider a graph with $10K$ nodes, $100K$ edges, and $99.9M$ disconnected (or negative) pairs. A (bad) model that ranks 1M false positives higher than the true edges achieves 0.99 AUC and 0.95 in AP under *biased testing* with equal negative samples.

Figures 4a and 4b show the receiver operating characteristic (ROC) and precision-recall (PR) curves for the model under *biased testing* with equal number of negative samples. Due to the downsampling, only 100k (out of 99.9M) negative pairs are included in the test set, among which only $100k/99.9M \times 1M \approx 1k$ pairs are ranked higher than the positive edges. In the ROC curve, this means that once the false positive rate reaches $1k/100k = 0.01$, the true positive rate would reach 1.0, leading to an AUC score of 0.99. Similarly, in the PR curve, when the recall reaches 1.0, the precision is $100k/(1k + 100k) \approx 0.99$, leading to an overall AP score of ~0.95.

By comparison, as shown in 4c, when the recall reaches 1.0, the precision under *unbiased testing* is only $100k/(1M + 100k) \approx 0.09$, leading to an AP score of ~0.05. This demonstrates that evaluation metrics based on *biased testing* provide an overly optimistic measurement of link prediction model performance compared to the more realistic *unbiased testing* setting.

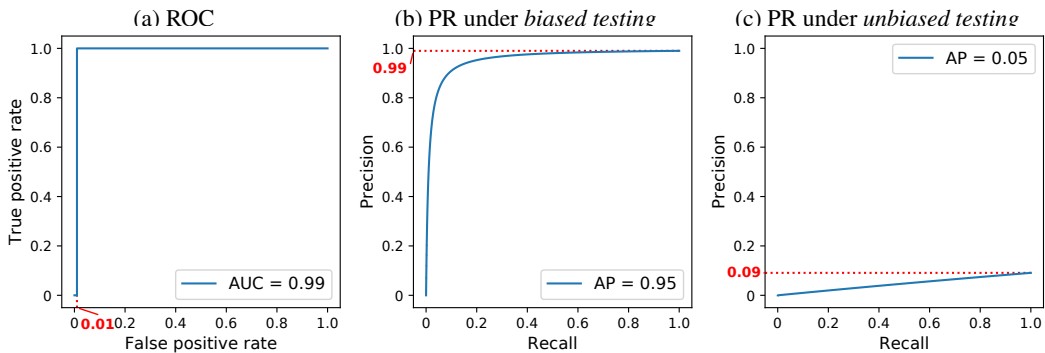

Figure 4: Receiver operating characteristic and precision-recall curves for the bad link prediction model that ranks 1M false positives higher than the 100k true edges. The model achieves 0.99 in AUC and 0.95 AP under *biased testing*, while the more informative performance evaluation metric, Average Precision (AP) under *unbiased testing*, is only 0.05.

## B    ABLATION STUDY

Here, we collect the results with the same hyperparameter setting as Gelato and present a comprehensive ablation study in Table 3. Specifically, *Gelato−MLP (AC)* represents Gelato without the MLP (Autocovariance) component, i.e., only using Autocovariance (MLP) for link prediction. *Gelato−NP (UT)* replaces the proposed N-pair loss (*unbiased training*) with the cross entropy loss (*biased training*) applied by the baselines. Finally, *Gelato−NP+UT* replaces both the loss and the training setting.

We observe that removing either MLP or Autocovariance leads to inferior performance, as the corresponding attribute or topology information would be missing. Further, to address the class imbalance problem of link prediction, both the N-pair loss and *unbiased training* are crucial for the effective training of Gelato.

We also present results for Gelato using different ranking-based loss functions. In particular, we choose between Precision@k, pairwise hinge, pairwise exponential, and pairwise logistic losses as candidates for replacing the N-pair loss based on Chen et al. (2009). The results are shown in Table 4, demonstrating that there is no clear winner considering the $hits@1000$ metric in the two datasets used (CORA and CITESEER).

Table 3: Results of the ablation study based on AP scores. Each component of Gelato plays an important role in enabling state-of-the-art link prediction performance.

|  | CORA | CITESEER | PUBMED |
|---|---|---|---|
| *Gelato−MLP* | $2.43 \pm 0.00$ | $2.65 \pm 0.00$ | $2.50 \pm 0.00$ |
| *Gelato−AC* | $1.94 \pm 0.18$ | $3.91 \pm 0.37$ | $0.83 \pm 0.05$ |
| *Gelato−NP+UT* | $2.98 \pm 0.20$ | $1.96 \pm 0.11$ | $2.35 \pm 0.24$ |
| *Gelato−NP* | $1.96 \pm 0.01$ | $1.77 \pm 0.20$ | $2.32 \pm 0.16$ |
| *Gelato−UT* | $3.07 \pm 0.01$ | $1.95 \pm 0.05$ | $2.52 \pm 0.09$ |
| *Gelato* | $\mathbf{3.90 \pm 0.03}$ | $\mathbf{4.55 \pm 0.02}$ | $\mathbf{2.88 \pm 0.09}$ |

Table 4: Comparison between N-pair loss (Gelato) against the Precision@K (PK), pairwise hinge (PH), pairwise exponential (PE), and pairwise logistic (PL) losses considering the $hits@1000$ metric.

|  | CORA | CITESEER |
|---|---|---|
| *Gelato-PK* | $16.32 \pm 0.19$ | $19.19 \pm 0.99$ |
| *Gelato-PH* | $\mathbf{18.09 \pm 0.48}$ | $16.56 \pm 0.13$ |
| *Gelato-PE* | $16.82 \pm 0.48$ | $15.9 \pm 0.34$ |
| *Gelato-PL* | $18.03 \pm 0.38$ | $17.14 \pm 0.66$ |
| *Gelato* | $16.62 \pm 0.31$ | $\mathbf{19.89 \pm 0.24}$ |

## C  DETAILED EXPERIMENT SETTINGS

**Positive masking.** For *unbiased training*, a trick similar to *negative injection* (Zhang and Chen, 2018) in *biased training* is needed to guarantee model generalizability. Specifically, we divide the training positive edges into batches and during the training with each batch $E_b$, we feed in only the residual edges $E - E_b$ as the structural information to the model. This setting simulates the testing phase, where the model is expected to predict edges without using their own connectivity information. We term this trick *positive masking*.

**Other implementation details.** We add self-loops to the enhanced adjacency matrix to ensure that each node has a valid transition probability distribution that is used in computing Autocovariance. The self-loops are added to all isolated nodes in the training graph for all datasets. Following the postprocessing of the Autocovariance matrix for embedding in Huang et al. (2021), we standardize Gelato similarity scores before computing the loss. We optimize our model with gradient descent via `autograd` in `pytorch` (Paszke et al., 2019). We find that the gradients are sometimes invalid when training our model (especially with the cross-entropy loss), and we address this by skipping the parameter updates for batches leading to invalid gradients. Finally, we use $prec@100\%$ on the (unbiased) validation set as the criteria for selecting the best model from all training epochs. The maximum number of epochs for CORA/CITESEER and OGBL-DDI/OGBL-COLLAB is set to be 100 and 250, respectively. For *partitioned testing*, we apply METIS (Karypis and Kumar, 1998) as our graph partitioning algorithm, due to its scalability and a balanced number of nodes per partition.

**Experiment environment.** We run our experiments in an *a2-highgpu-1g* node of the Google Cloud Compute Engine. It has one NVIDIA A100 GPU with 40GB HBM2 GPU memory and 12 Intel Xeon Scalable Processor (Cascade Lake) 2nd Generation vCPUs with 85GB memory.

**Reference of baselines.** We list link prediction baselines and their reference repositories we use in our experiments in Table 5. Note that we had to implement the batched training and testing for several baselines as their original implementations do not scale to *unbiased training* and *unbiased testing* without downsampling.

Table 5: Reference of baseline code repositories.

| Baseline | Repository |
|---|---|
| SEAL (Zhang and Chen, 2018) | `https://github.com/facebookresearch/SEAL_OGB` |
| Neo-GNN (Yun et al., 2021) | `https://github.com/seongjunyun/Neo-GNNs` |
| BUDDY (Chamberlain et al., 2023) | `https://github.com/melifluos/subgraph-sketching` |
| NCN / NCNC (Wang et al., 2023) | `https://github.com/zexihuang/random-walk-embedding` |

## D    LINK PREDICTION ADDITIONAL RESULTS

Table 6: Link prediction performance comparison (mean ± std AP) for all datasets considered. Gelato consistently outperforms GNN-based methods, topological heuristics, and two-stage approaches combining attributes/topology, being at least in the top-3 best-performing models in all datasets. For CORA, CITESEER, OGBL-DDI and PUBMED results we used *unbiased* training, while for OGBL-COLLAB *partitioned* sampling is used, for scalability reasons. The top three models are colored by **First**, **Second** and **Third**.

|  |  | CORA | CITESEER | PUBMED | OGBL-DDI | OGBL-COLLAB |
|---|---|---|---|---|---|---|
| GNN | SEAL | 2.21$^*$ | 2.43$^*$ | *** | 35.2$^*$ | **47.43**$^*$ |
|  | Neo-GNN | 2.15 ± 1.51 | 1.71 ± 0.06 | 1.21 ± 0.14 | 24.42$^*$ | 31.86$^*$ |
|  | BUDDY | 1.20 ± 0.25 | 1.72 ± 0.08 | OOM | 21.59 ± 1.02 | **47.13 ± 0.22** |
|  | NCN | 1.82 ± 0.49 | **2.79 ± 0.21** | 0.06 ± 0.07 | **44.75 ± 0.07** | 41.38 ± 0.44 |
|  | NCNC | **2.88 ± 0.16** | **3.23 ± 0.44** | **1.54 ± 0.01** | **44.9 ± 0.05** | 27.67 ± 3.3 |
| Topological Heuristics | CN | 1.10 ± 0.00 | 0.74 ± 0.00 | 0.36 ± 0.00 | 24.76 ± 0.00 | 24.18 ± 0.00 |
|  | AA | 2.07 ± 0.00 | 1.24 ± 0.00 | **2.50 ± 0.00** | 25.25 ± 0.00 | 34.28 ± 0.00 |
|  | AC | **2.43 ± 0.00** | 2.65 ± 0.00 | **2.50 ± 0.00** | **29.42 ± 0.00** | 37.92 ± 0.00 |
| Gelato |  | **3.90 ± 0.03** | **4.55 ± 0.02** | **2.88 ± 0.00** | **29.42 ± 0.00** | 42.53$^*$ |

$^*$ Run only once as each run takes >24 hrs.

Table 7: Link prediction performance comparison (mean ± std MRR). Gelato shows competitive performance, despite its simplicity, being in the top-3 best-performing models in almost all datasets. We highlight that Gelato is the best-performing method in PUBMED and OGBL-COLLAB, the hardest evaluation regimes since we consider the *unbiased testing* scenario for both datasets. The top three models are colored by **First**, **Second** and **Third**.

|  |  | CORA | CITESEER | PUBMED | OGBL-DDI | OGBL-COLLAB |
|---|---|---|---|---|---|---|
| GNN | SEAL | 0.0204$^*$ | 0.235$^*$ | *** | 0.0071$^*$ | **4.9441**$^*$ |
|  | Neo-GNN | 0.2216 ± 0.101 | 0.0969 ± 0.0285 | 0.0001 ± 0.0001 | **0.0098**$^*$ | 0.3435$^*$ |
|  | BUDDY | 0.136 ± 0.0607 | 0.121 ± 0.0026 | OOM | 0.0094 ± 0.0003 | **1.2285 ± 0.0576** |
|  | NCN | 0.1216 ± 0.0551 | **0.1989 ± 0.0515** | **0.0005 ± 0.0007** | **0.0117 ± 0.002** | 0.1343 ± 0.0588 |
|  | NCNC | **0.4606 ± 0.1867** | **0.2934 ± 0.1746** | 0.0002 ± 0.00004 | **0.0171 ± 0.0133** | 0.011 ± 0.0042 |
| Topological Heuristics | CN | 0.1816 ± 0.00 | 0.0933 ± 0.00 | 0.0001 ± 0.0000 | 0.0103 ± 0.00 | 0.4767 ± 0.00 |
|  | AA | 0.1764 ± 0.00 | 0.1154 ± 0.00 | 0.0001 ± 0.0000 | 0.0104 ± 0.00 | 0.0333 ± 0.00 |
|  | AC | **0.3069 ± 0.00** | 0.1245 ± 0.00 | **0.0006 ± 0.00** | 0.0084 ± 0.00 | 0.7692 ± 0.00 |
| Gelato |  | **0.2558 ± 0.0001** | **0.1424 ± 0.0028** | **0.0009 ± 0.0003** | 0.0084 ± 0.001 | **6.1422**$^*$ |

$^*$ Run only once as each run takes >24 hrs.

## E    GELATO - UNBIASED VS PARTITIONED

Figure 5 shows *partitioned sampling* is a good proxy to obtain splits that are both realistic and scalable through the evolution of the $hits@1000$ metric evaluated on *partitioned test* on Cora. It is possible to verify that there is almost no performance performance gap between *partitioned* and *unbiased* training.

## F    ADDITIONAL LINK PREDICTION MODEL COMPARISON

Despite its simplicity, Gelato is consistently among the best link prediction models considering $hits@k$ and $prec@k$. We demonstrate the competitive results of Gelato against the GNN-based models by varying $k$ in Figures 6 and 7.

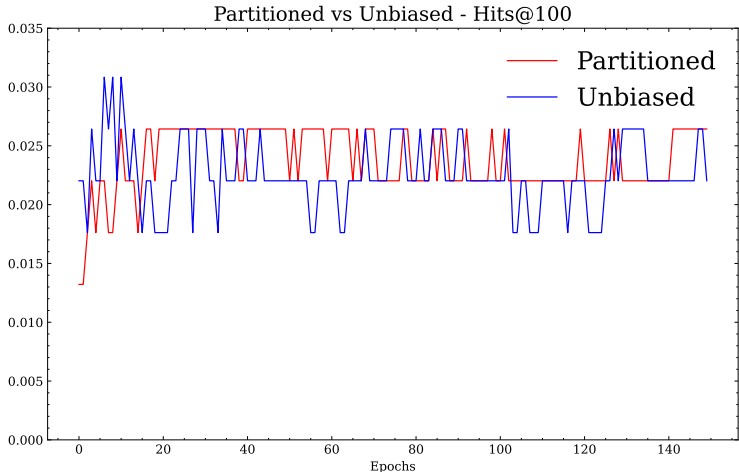

Figure 5: Comparison between Gelato trained using *unbiased* sampling against *partitioned* sampling on Cora ($p = 10$, $\approx 250$ nodes per partition), in which we verify that there's almost no performance gap between both models, but the *partitioned* sampling approach trains 2.8x times faster than the *unbiased* sampling approach. The speedup increases with the number of partitions.

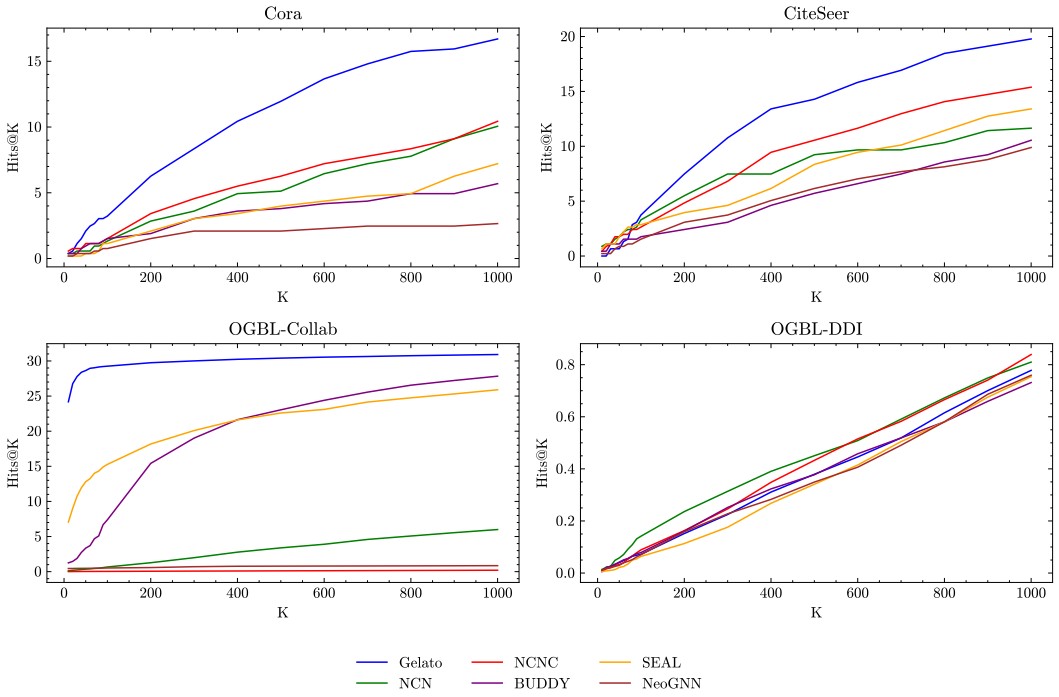

Figure 6: Link prediction comparison in terms of $hits@k$ using Cora, CiteSeer, OGBL-DDI and OGBL-Collab. All datasets were split using *unbiased* sampling, except OGBL-Collab, which was split using *partitioned* sampling. Gelato obtains the best performance on Cora and OGBL-Collab by a large margin and remains competitive on CiteSeer and OGBL-DDI, a dataset in which all methods struggle.

## G  NON-NORMALIZED PARTITIONED SAMPLING RESULTS

We recreate Figure 3 with non-normalized densities to show the extreme difference in the number of negative and positive pairs.

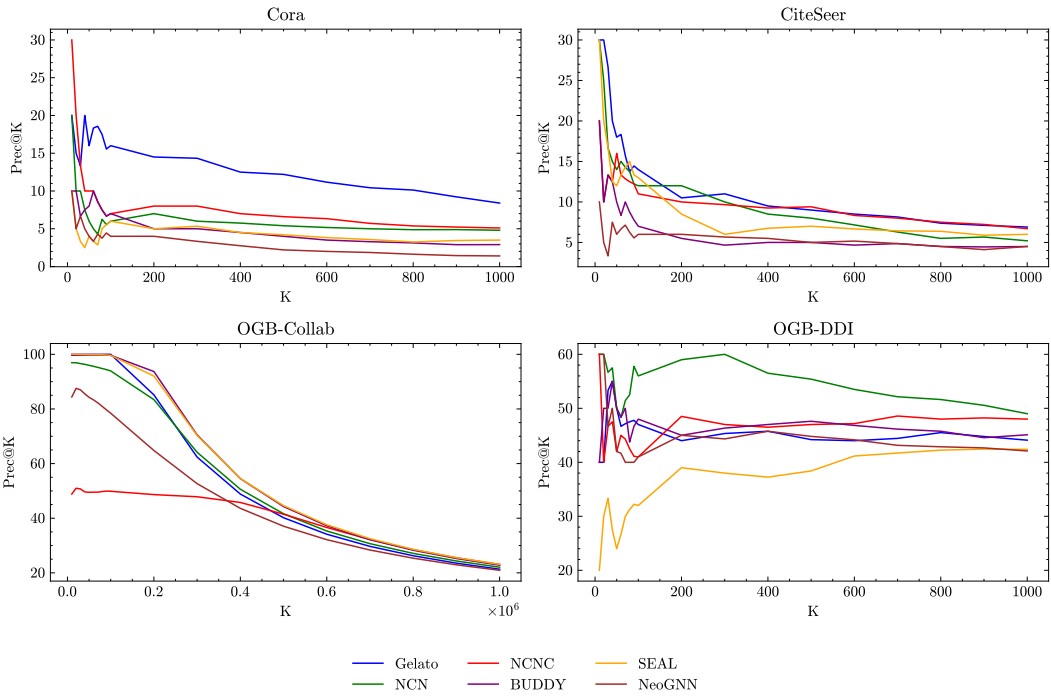

Figure 7: Link prediction comparison in terms of *prec@k* using Cora, CiteSeer, OGBL-DDI and OGBL-Collab. All datasets were split using *unbiased* sampling, except OGBL-Collab, which was split using *partitioned* sampling. Gelato obtains the best performance on Cora and OGBL-Collab by a large margin and remains competitive on CiteSeer and OGBL-DDI, a dataset in which all methods struggle.

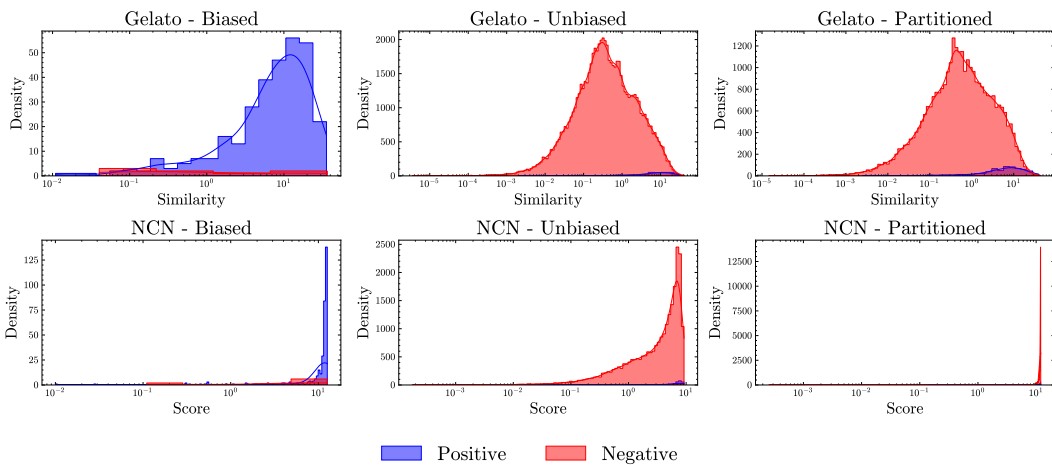

Figure 8: The non-normalized version of the Figure 3. Negative pairs are represented in red, and positive pairs are represented in blue. For unbiased and partitioned testing, negative pairs are significantly more likely than positive ones—due to graph sparsity—even for the largest values of similarity or scores. For this reason, for any decision boundary chosen, distinguishing positive pairs from negative ones is like finding "needles in a haystack".

## H  Time comparison

In Table 8 we compare the time per epoch between Gelato and our two main competitors: BUDDY and NCN (the faster version of NCNC). It is possible to notice a few patterns: Gelato suffers with

graphs with a very large number of nodes, whereas NCN gets worse results in denser networks (due to the CommonNeighbors dependency), despite being the fastest. Buddy relies on storing hashes, which results in an OOM error when running PubMed on the *unbiased training* scenario.

Table 8: Estimated time (in seconds) per epoch.

|  | BUDDY | NCN | Gelato |
|---|---|---|---|
| Cora | 0.4s | 3.5s | 3s |
| CiteSeer | 926s | 4.6s | 2.6s |
| PubMed | OOM | 5.0s | 49s |
| OGBL-DDI | 720s | 30s | 0.5s |
| OGBL-Collab | 127s | 21s | 5400s |

## I  BIASED TRAINING RESULTS

We present results for Gelato trained and evaluated in the *unbiased / partitioned* test in Table 9 for the small datasets. The results show a performance degradation for most models in almost all datasets, especially for BUDDY and NCN. SEAL, NeoGNN, and Gelato have better robustness obtaining even better results comparatively in some scenarios.

## J  CLUSTERING TIMES

We chose METISKarypis and Kumar (1998) as our graph partitioning method due to its scalability and the fact it produces partitions with a similar number of nodes. METIS runs as a pre-processing step in our pipeline to enable *partitioned* sampling, in which we consider only negative pairs within each partition. We display in Table 10 the clustering time for each dataset and the number of partitions considered using the METIS implementation available in the torch-sparse (`https://github.com/rusty1s/pytorch_sparse`) Python package.

## K  GNN RESULTS

We substitute the MLP module of Gelato with a GNN module using GIN Xu et al. (2018) (GelatoGIN). The results are displayed in 9, depicting an overfitting scenario that is more pronounced in GelatoGIN considering $prec@k$ results.

## L  SENSITIVITY ANALYSIS AND LEARNING HYPERPARAMETERS

We conduct a sensitivity analysis of both $\alpha$ and $\beta$ hyperparameters considering $AP$ on validation as the reported metric, reported in Figure 10. The other two hyperparameters are set to $\eta = 0$ and $T = 3$ in both scenarios. We show that there is a smooth transition between the values of AP obtained through different hyperparameters, facilitating hyperparameter search.

Table 9: We show present results (mean $\pm$ std MRR) for GNN methods versus Gelato trained in *biased training* splits and evaluated on *unbiased* (Cora and CiteSeer) and *partitioned* (DDI) splits. The top three models are colored by **First**, **Second** and **Third**.

|  |  | CORA | CITESEER | OGBL-DDI |
|---|---|---|---|---|
| GNN | SEAL | **0.0637**[*] | **0.1602**[*] | **0.0067**[*] |
|  | NeoGNN | 0.0505 ± 0.0145 | **0.2312 ± 0.0796** | **0.0544**[*] |
|  | BUDDY | **0.1776 ± 0.0** | 0.12 ± 0.0 | 0.0059 ± 0.0 |
|  | NCN | 0.0012 ± 0.0012 | 0.0126 ± 0.0176 | 0.0006 ± 0.0001 |
| | Gelato | **0.2564 ± 0.0069** | **0.1418 ± 0.0011** | **0.0084 ± 0.0** |

[*] Run only once as each run takes >24 hrs.

Table 10: METIS clustering time for each dataset in seconds. METIS executes scalable and fast graph partitioning adding negligible running time to the pre-processing step.

|  | # Partitions | Time (s) |
|---|---|---|
| Cora | 10 | 0.07 |
| CiteSeer | 10 | 0.03 |
| PubMed | 100 | 0.16 |
| OGBL-DDI | 20 | 0.42 |
| OGBL-Collab | 1300 | 1.91 |

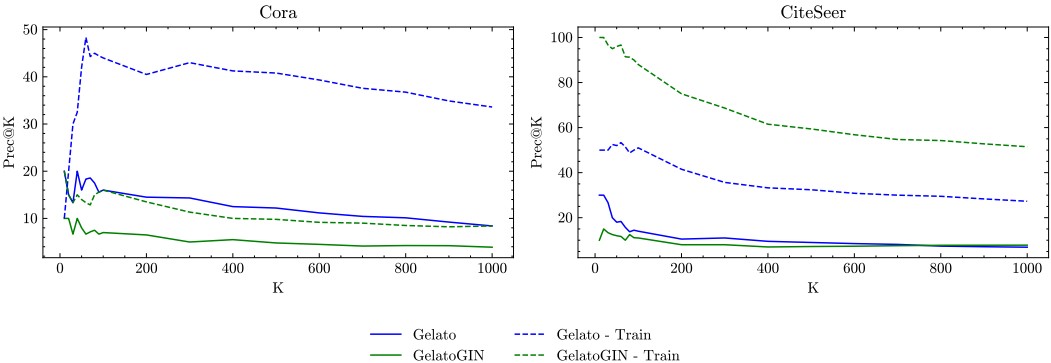

Figure 9: Performance comparison ($prec@k$) between Gelato (in **blue**) against GelatoGIN (in **green**), which replaces the MLP module by GIN. The dashed line represents the performance on training, while the full line represents the performance on test. We can see that despite eventually obtaining better results on training (CiteSeer), this performance is not matched by the test results, demonstrating overfitting.

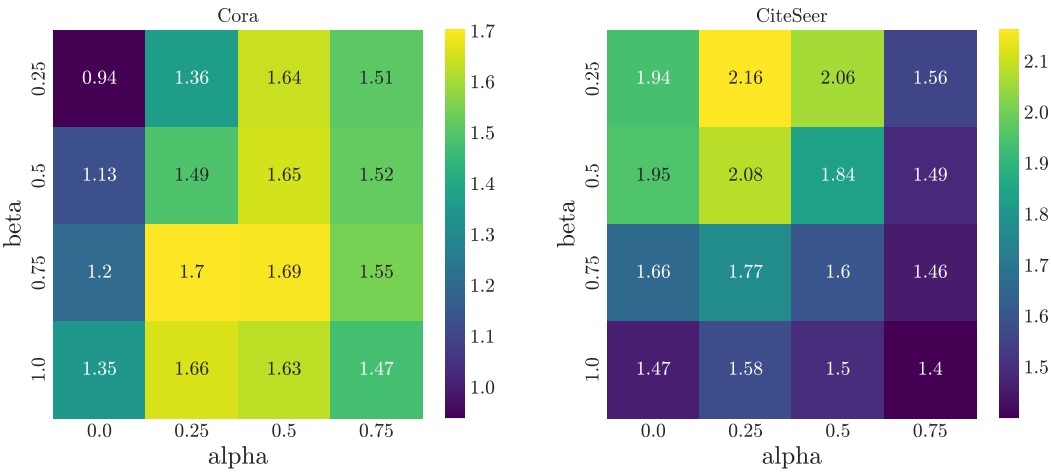

Figure 10: Sensitivity analysis of $\alpha$ and $\beta$ considering $AP$ metric. .

We also present in Figure 11 results treating both $\alpha$ and $\beta$ as learnable parameters, showing that this procedure does not improve the $prec@k$ or $hits@k$ results. The values found for the hyperparameters were $\alpha = 0.5670$ and $\beta = 0.4694$ on Cora and $\alpha = 0.5507$ and $\beta = 0.4555$ on CiteSeer.

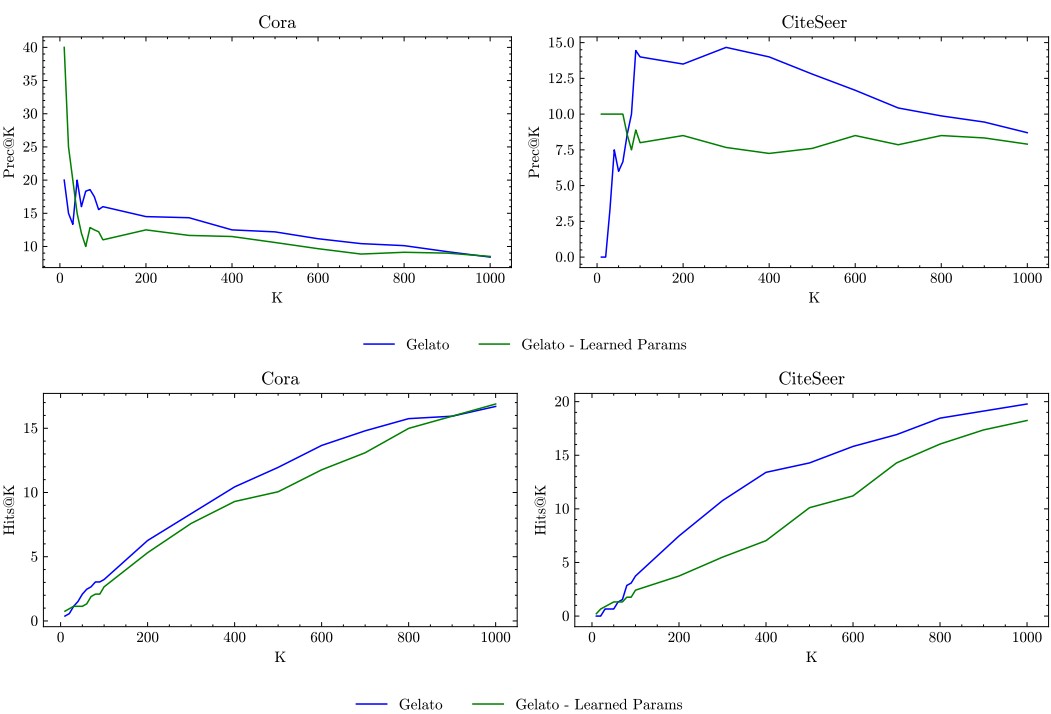

Figure 11: Results of $prec@k$ (top) and $hits@k$ (bottom) of Gelato (in (in **blue**)) against Gelato with $\alpha$ and $\beta$ as learning parameters (in **green**). In both datasets and metrics considered, the learned $\alpha$ and $\beta$ obtained worse values than the values found by the grid search hyperparameter tuning strategy.

