# OpenReview forum: "Attribute-Enhanced Similarity Ranking for Sparse Link Prediction"
_ICLR.cc/2024/Conference — Submitted to ICLR 2024_

### Official Review · Reviewer_4LRj · 2023-10-27

**Soundness:** 3 good
**Presentation:** 3 good
**Contribution:** 2 fair
**Rating:** 5
**Confidence:** 4

**Summary:**

The authors introduce Gelato - “Graph enhancement for link prediction with autocovariance” -, a new similarity-based link prediction algorithm enhancing an existing topological heuristic (Autocovariance) with graph learning, specially designed to handle very sparse graphs.

This approach, that leverages the node attribute information, predicts edges between node pairs with high Autocovariance similarity. The parameters of the MLP model used to incorporate node attribute information into the weight of the edge are optimized through a ranking-based N-pair loss for which hard training negative node pairs are sampled.

**Strengths:**

The paper is well written and quite enjoyable to read. The authors are well explaining how certain state-of-the-art methods such as graph neural networks (GNN) are actually tested in a biased manner (not including all negative pairs, in potentially non-sparse graphs) where in practice graphs are usually sparse and not all positive pairs are known.

They also highlights performance limitations of GNN: a binary classification loss sensitive to class imbalance and the inability to learn strong topology of the data to better introduce Gelato.

Steps of the algorithm are clearly stated:

-The node attribution information is used to extend the set of initial edges with new ones between nodes for which similarity is higher than a specified threshold.

-The new augmented graph is associated with a trainable weight which is learned via a MLP network. The final edge weights are a weighted combination of the topological, MLP-learned weights and similarity score between the node features. By design, the final graph contains both structural and attribute information.

-To this enhanced graph, we associate negative pairs which are generated within node partitions obtained via the METIS algorithm (I wish there was more explanation though, not to describe METIS algorithm itself but how sampling is performed from the multiple hiearchical partitions). The objective of this process is to have hard negative pairs for the training.

-Then the trainable weights of the MLP model - assumed to learn meaningful interactions between any two node attributes linked by an edge in the enhanced graph- are optimized so that to rank the positive edges between 2 nodes higher than the negative ones (instead of classifying them between positive and negative ones which is complex for very unbalanced classes).

Focusing on accuracy AND scalability, it is indeed discussed how to efficiently compute the n-square Autocovariance similarity matrix (used to define the weighted adjacency matrix of the enhanced graph).

Code of the approach is made publicly available, which is highly appreciated for reproducibility of the results and wide adoption of the approach by the ML community.

Experiments are made on 4 public real-world datasets and Gelato is compared against 4 SOTA GNN-based approaches as well as 3 heuristics.
They show that Gelato is outperforming GNN-based approaches in the unbiased setting regarding link prediction performance (2nd best approach is the heuristic that Gelato is empowering with graph learning).

However, in the bias and partitioned settings, NCN seems to be better at distinguishing positive pairs from the negative ones.

**Weaknesses:**

In METIS, the original graph is transformed into sequentially smaller graphs $G_1$,$G_2$, …, $G_p$ such that $|V_0| > |V_1| > |V_2| >...> |V_p|$. What is unclear to me is how exactly the negative edges are sampled from the hierarchical partitions. Maybe some pictures to describe the process might be helpful.

As highlighted by the authors, in the partitioned setting, NCN seems to be better at distinguishing positive pairs from the negative ones compared to Gelato.

Potential limitation of the paper is that the contribution is mostly experimental (from sometimes existing blocks) and as a consequence we might expect more experiments to support the proposed approach (GNN instead of MLP in Gelato is left for future work for instance, same as for other types of ranking-based losses that could have been compared).

**Questions:**

Q1: Can you explain a little more in detail how you generate negative edges from the partitions? Do you have p ways of generating negative edges? In comparison is unbiased sampling corresponding to sampling ALL possible negative edges?

Q2: Is Gelato also outperforming other SOTA algorithms for non-sparse graphs, in the described biased setting?

Q3: Did you investigate the learning of the \alpha and \beta hyperparameters?

=== AFTER REBUTTAL ===

Many thanks to the authors for taking the time to answer my questions. Thank you very much for the extra experiments that help to get a sense of how Gelato is behaving. However, I wish to keep my overall score.

---

> ### Author Response · Authors · 2023-11-23
> **Response to reviewer 4LRj**
>
> We thank the reviewer for the comments. The reviewer was able to identify some of the strengths of our paper and seems to share some of our excitement regarding this work. We were able to address each of them and hope that the reviewer would consider upgrading the rating based on our responses:
>
> __1. Negative sampling based on partitions.__
>
> We apply the final (lowest-level) partitions for negative sampling. Negative pairs across partitions are not sampled during training. A more complex sampling scheme based on the entire hierarchy generated by METIS could be effective but we found that our simple approach already efficiently approximates results from unbiased training (where all negative pairs are used for training), as shown in Figure 3. We will clarify this in our paper.
>
> __2. NCN vs Gelato.__
>
> The results from Figure 3 show that NCN is unable to distinguish between positive and negative pairs in unbiased and partitioned settings (negative and positive pairs have very similar values of edge probability). This limitation is reflected in the results from Table 2, where Gelato significantly outperforms NCN in 4/5  datasets. NCN is only competitive in the biased testing setting.
>
> __3. Contributions and additional experiments.__
>
> We agree with the reviewer and have added several additional results to the Appendix of our paper, including a sensitivity analysis, an ablation study, running times, and multiple evaluation metrics. We have implemented the GNN version of our approach, but the results have shown no benefits in accuracy due to overfitting. We have also included the results of Gelato with four alternative ranking-based losses (see Appendix B) and there is no clear winner among the alternatives. We will include these results in the updated version of our paper.
>
>  __4. Gelato vs SOTA for non-sparse graphs in the biased setting.__
>
> Some of the main contributions of Gelato, namely the topological heuristic, negative sampling, and n-pair loss are motivated by sparse graphs. Our results show that Gelato consistently outperforms the baselines for sparse graphs. We note that most relevant graphs in real-world applications are expected to be sparse [1,2]. Table 1 shows that 4/5 graphs considered in our experiments are sparse.
>
> *[1] Albert-Lászlo  Barabasi. Network Science. Cambridge University Press, 2016.*
>
> *[2] Mark Newman. Networks. Oxford University Press, 2018.*
>
> When evaluated using unbiased testing, most methods (Gelato and the baselines) benefit significantly from unbiased training and partition-based negative sampling. Appendix I shows results for all the methods using biased training. We note that BUDDY and NCN in particular suffer the most with overfitting in the biased training scenario, while SEAL, NeoGNN, and Gelato obtain the most robust results.
>
> __5. \alpha and \beta hyperparameters.__
>
> We have added a sensitivity analysis and experiments learning \alpha and \beta in Appendix L. The results show a smooth transition between AP values obtained by different hyperparameter configurations, enabling easier application of tunning techniques such as grid search and AutoML. Further, treating \alpha and \beta as learnable parameters does not show improvements either in prec@k or in hits@k (as shown in Figure 11).

---

### Official Review · Reviewer_554z · 2023-10-31

**Soundness:** 2 fair
**Presentation:** 3 good
**Contribution:** 2 fair
**Rating:** 3
**Confidence:** 4

**Summary:**

This paper studies link prediction on attributed graphs. It argues that a balanced number of disconnected pairs does not translate to the more realistic imbalanced setting and proposes some techniques to develop Gelato, with a graph learning based on node attributes, a ranking loss for handling class imbalance, and a negative sampling technique to handle hard training pairs.

**Strengths:**

1.	The paper is easy to follow with reasonable clarity to understand the techniques.
2.	Experiments are conducted on several benchmark datasets.

**Weaknesses:**

1.	The techniques in the proposed method in Section 3.1 to 3.4 are mostly existing techniques simply adopted into the paper. The novelty of the proposed method is quite unclear. Many attributed graph representation learning methods exist.
2.	The unbiased setting is not well-motivated. Why consider all random node pairs in a graph? For disconnected pairs, given a node, you can just sample the node pairs near the node via graph topology, e.g., within 2-hop, 3-hop. I do not agree that random sampling of all node pairs is more realistic.
3.	Though the experiments are conducted on real datasets, the results cannot reveal too much about the benefits brought by the proposed method. Also as mentioned above, the techniques of attributed graph learning in Section 3.1, 3.2, and N-pair loss in 3.3, negative sampling in 3.4 are with unclear novelty as they are mostly existing techniques.

**Questions:**

Please see weaknesses.

---

> ### Author Response · Authors · 2023-11-23
> **Response to reviewer 554z**
>
> We thank the reviewer for the comments. We were able to address each of them and hope that the reviewer would consider upgrading the rating based on our responses:
>
> __1. Novelty of our paper.__
>
> Our paper is focused on the link prediction problem and our solution differs significantly from the state-of-the-art because (1) it enhances a topological heuristic instead of applying graph neural networks, (2) it proposes a partition-based negative sampling scheme instead of biased sampling, and (3) it applies the n-pair loss instead of the cross-entropy. Recent work on link prediction in the last five years has attempted to improve SEAL (the first GNN for link prediction) [1] while still applying GNNs, biased sampling, and cross-entropy, thus we claim that our solution while simple is still more novel than most recent papers on supervised link prediction. Finally, we highlight that despite the simplicity of our approach, it improves over BUDDY (accepted as notable-top-5% at ICLR’23)  and NCNC (under review for ICLR’24), two bigger and more complex baselines.
>
> *[1] Muhan Zhang and Yixin Chen. Link prediction based on graph neural networks. In NeurIPS, 2018.*
>
> __2. The unbiased setting is not well-motivated. Why consider all random node pairs in a graph? For disconnected pairs, given a node, you can just sample the node pairs near the node via graph topology, e.g., within 2-hop, 3-hop. I do not agree that random sampling of all node pairs is more realistic.__
>
> We note that existing approaches (biased) also sample negative pairs at random. We use unbiased sampling to show the limitations of biased sampling, which is applied by recent approaches for link prediction. The link prediction problem as proposed originally does not make any assumption about which pairs are potential links based on topological distance  [1,2]. However, we show that Gelato can be trained with within-partition node pairs (this is not very different from what the reviewer has proposed) while still being able to predict links between any pair during testing. In fact, Figure 3 shows that our approach is an efficient approximation to unbiased sampling.
>
> *[1] David Liben-Nowell and Jon Kleinberg. The link-prediction problem for social networks. Journal of the American society for information science and technology, 58(7):1019–1031, 2007.*
>
> *[2] Linyuan Lu and Tao Zhou. Link prediction in complex networks: A survey. Physica A: statistical mechanics and its applications, 390(6):1150–1170, 2011.*
>
> __3. Benefits brought by the proposed methods.__
>
> As with most machine learning solutions, the benefits of link prediction methods are often shown via experiments. That has led to the popularization of graph benchmarks such as Imagenet for computer vision and OGB [1] for graphs.  Our approach combines building blocks (topological heuristic, graph learning, negative sampling, and n-pair loss) proposed earlier in the literature in an innovative fashion as they are not applied by existing link prediction methods.
>
> *[1] Hu et al. Open graph benchmark: Datasets for machine learning on graphs. In NeurIPS, 2020.*

---

### Official Review · Reviewer_6oyf · 2023-10-31

**Soundness:** 1 poor
**Presentation:** 2 fair
**Contribution:** 1 poor
**Rating:** 3
**Confidence:** 4

**Summary:**

The paper proposes a method for link prediction in sparse graph to solve the so-called problem of "biased testing" that select a small portion of negative pair in evaluation of the model

**Strengths:**

N/A

**Weaknesses:**

The chief problem the paper raised was the small portion of negative pair used in testing. The arguments do not back up the claim.
- These balanced sampling methods overestimate the ratio of positive pairs. There is no evidence that any of these methods estimating ratios of positive pairs.
- "AUC is not an effective evaluation metric for link prediction as it is biased towards the majority class". This needs an evidence!
- The example show that negative pairs have <2% of intra-block pairs. What is the problem with this? In fact, one can split any negative sample set into any two imbalanced subsets, then the smaller subset will likely have a smaller number in the sampled training data. Is that smaller subset a "hard" set and, just by splitting, should this dataset have "biased testing" as well?

The paper can call this problem any name, but not "biased testing" as "bias" is a establish statistical term.

**Questions:**

N/A

---

> ### Author Response · Authors · 2023-11-23
> **Response to reviewer 6oyf**
>
> We thank the reviewer for the comments. We were able to address each of the comments, as they were mostly clarification questions, and hope that the reviewer would consider upgrading the rating based on our responses:
>
> __1. These balanced sampling methods overestimate the ratio of positive pairs. There is no evidence that any of these methods estimate ratios of positive pairs.__
>
> As discussed in Section 5, existing GNNs for link prediction formulate the problem as a binary classification, where a sigmoid function converts model outputs into edge probabilities. However, because of biased training, these outputs will also be biased, overestimating the ratio of positive pairs (edge probabilities will be similar to non-edge ones). In sparse graphs, the edge probabilities tend to be very small as illustrated in Table 1. We will clarify this in the updated version of the paper.
>
> __2. Ineffectiveness of AUC in imbalanced settings.__
>
> We have added a detailed explanation of this phenomenon in Appendix A. Consider a graph with 10K nodes, 100K edges, and 99.9M disconnected (or negative) pairs. A (bad) model that ranks 1M false positives higher than the true edges achieves 0.99 AUC and 0.95 in AP under biased testing with equal negative samples. We have also added ROC and PR curves supporting our claim. This behavior has also been highlighted by previous work  [1,2].
>
> *[1] Davis and Goadrich. The relationship between precision-recall and roc curves. In ICML, 2006.*
>
> *[2] Saito and Rehmsmeier. The precision-recall plot is more informative than the roc plot when evaluating binary classifiers on imbalanced datasets. PloS one, 10(3):e0118432, 2015*
>
> **3. The problem with less than 2% of negative edges being intra-block pairs.**
>
> We have illustrated the impact of biased sampling in Figure 3. Methods based on biased sampling are unable to distinguish positive and negative pairs within clusters because they are trained with significantly more “easy” negative pairs across clusters. To be more specific on the bias here, consider all node pairs (including both positive ones and negative ones) as a single population. We are interested in estimating the probability of drawing positive edges from this population. Sampling only an equal number of negative edges as the total positive edges result in a biased estimation of this probability, even though the uniform sampling with respect to the negative edges themselves is unbiased.
>
> **4. The use of the term "bias".**
>
> See the answer above. In biased training and testing node pairs are not selected uniformly but positive pairs have a much higher chance of being selected. This is consistent with the definition of bias in statistics.

---

### Official Review · Reviewer_LQFc · 2023-10-31

**Soundness:** 2 fair
**Presentation:** 3 good
**Contribution:** 2 fair
**Rating:** 3
**Confidence:** 4

**Summary:**

This paper rethinks the disadvantages of most current link prediction techniques for sparse graphs. Specifically, for sparse graphs, since most negative pair samples are quite different from the positive ones, metrics that sample only a few negative pairs lead to biased testing. To address this problem, this paper suggests that unbiased testing exploiting all negative samples is more persuasive, especially for sparse graphs, and even finds that unsupervised topological heuristics (Autocovariance) can outperform most advanced link prediction techniques. Under this perspective, this paper devises a novel link prediction approach, called Gelato, that is suitable for sparse graphs. Gelato contains four parts: 1) graph learning: incorporate attribute similarity, topological weights, the untrained weights, and the trained weights into the new adjacency matrix; 2) topological heuristic: borrow the heuristic from AC and use equation 6 to alleviate the memory overhead; 3) N-pair loss: relieve the sensitivity to class imbalance; 4) negative sampling: use graph partitioning to generate hard negative samples.

**Strengths:**

+ Considering that various negative pair samples are actually far from the real links, the motivation that targets unbiased testing is reasonable and the idea makes sense.
+ The writing is clear to show all the details of the proposed Gelato.
+ For the metric hit@1000, Gelato outperforms other baselines on most benchmarks, especially the sparse graphs, which empirically proves the validity of Gelato on sparse graphs.

**Weaknesses:**

- The design that includes attribute similarity, topological weights, the untrained weights, and the trained weights incorporates three hyper-parameters $\epsilon_{\eta}$, $\alpha$, and $\beta$ to control the weights, which makes it more difficult to tune an optimal model.
- Though different parts are delicately devised to improve the performance of Gelato, the ablation study is missing to show the actual effect. For example, does N-pair loss really work, and is better than CE? When ignoring the attribute similarity, will the performance of Gelato still outperform the one of AC? This is quite important because Gelato works worse than AC on dataset OGBL-DDI, which is the only dataset containing no natural node features.
- If possible, the performance of Gelato with different hyperparameters is better to display, which can also show the effect of different modules in Gelato.
- For non-sparse graphs, Gelato seems not to be superior.

**Questions:**

- How can the model be trained in an end-to-end manner considering that the negative samples need to be partitioned?
- The average degree of OGBL-DDI seems to be wrong.
- Can the authors explain the influence and application of link prediction considering that only a few samples are hit in all the 1,000 samples? Is there any industrial effect when improving the hits@1000 from 10 to 20?
- Since the performance of Gelato is not significant on OGBL-DDI and this dataset is the densest one, can I put forward the conclusion that Gelato is only suitable and useful for sparse graphs?
- What is the time overhead of graph partitioning? Will it occupy much time in the whole model?

---

> ### Author Response · Authors · 2023-11-23
> **Response to reviewer LQFc**
>
> We thank the reviewer for the comments. We were able to address each of them and hope that the reviewer would consider upgrading the rating based on our responses:
>
> **1. Difficulty in tuning an optimal model.**
>
> We have included a sensitivity analysis of the hyperparameters in Appendix L. The results show that there is a smooth transition between the values of AP obtained through different hyperparameters, enabling easier hyperparameter tunning through strategies such as grid search and AutoML.
>
> **2. Ablation study.**
>
> We have included an ablation study in Appendix B. The results show that each component of Gelato contributes to its accuracy results.
>
> **3. Gelato’s performance on non-sparse graphs.**
>
> Some of the main contributions of Gelato, namely the topological heuristic, negative sampling, and n-pair loss are motivated by sparse graphs. We note that most relevant graphs in real-world applications are expected to be sparse [1,2]. Table 1 shows that 4/5 graphs considered in our experiments are sparse.
>
> *[1] Albert-Lászlo  Barabasi. Network Science. Cambridge University Press, 2016.*
>
> *[2] Mark Newman. Networks. Oxford University Press, 2018*
>
> **4. How can the model be trained in an end-to-end manner considering that the negative samples need to be partitioned?**
>
> The partitioning step is performed as a preprocessing (similar to ClusterGCN [1]). In summary, after partitioning the graph, we apply the unbiased sampling procedure within each partition, which enables sampling harder pairs, while avoiding the high memory cost of naively applying unbiased sampling in the entire graph (|V|^2). We will clarify this in the updated version of the paper.
>
> *[1] Chiang et al. Cluster-gcn: An efficient algorithm for training deep and large graph convolutional networks. KDD’19.*
>
> **5. The average degree of OGBL-DDI seems to be wrong.**
>
> The average degree (500.5) reported in our paper is the same as the one reported in the original OGB paper: https://cs.stanford.edu/people/jure/pubs/ogb-neurips20.pdf (see Table 2).
>
> **6. The industrial effect when improving the hits@1000 from 10 to 20.**
>
> The value of hits@1000 can be misleading as each edge is ranked against every negative pair (a large number for sparse graphs). In Appendix D, we report the results in terms of average precision, and, in Appendix F, we report results for precision@k. For instance, on Cora, about 10% of the top 1000 links predicted by our approach are correct.
>
> **7. Overhead of graph partitioning.**
>
> Graph partitioning is performed as a preprocessing step and can be done efficiently using METIS, which can scale to large graphs. We have added the partitioning time in Table 9, showing that the overhead is negligible (in the order of seconds).

---

### Author Response · Authors · 2023-11-23
**Response to all reviewers**

We thank all the reviewers for their comments and for the opportunity to address the raised concerns in this rebuttal. Most reviewers are satisfied with the presentation of our paper but the concerns are the soundness and novelty. We have worked intensively during this rebuttal period to address each of the raised concerns to the best of our ability, including clarifications and new experiments (sensitivity analysis, ablation study, etc.).

While the improvement in soundness due to the rebuttal might be more evident, we want to emphasize the contributions of our work. The reviewers (especially __554z__ and __4LRj__) have noted that attributed link prediction and the main components of Gelato (topological heuristic, graph learning, negative sampling, and n-pair loss) have been proposed earlier. However, we want to note that the 13 papers on supervised link prediction published in the last seven years at top machine learning conferences, including ICLR and NeurIPS, proposed variations of GNNs for link prediction. None of them shares the contributions of Gelato and our results provide strong evidence that Gelato is a better alternative than these recent approaches for sparse graphs. In particular, Gelato achieves favorable results compared with BUDDY (accepted as notable-top-5% at ICLR’23)  and NCNC (under review for ICLR’24). Table 2 shows that our approach outperforms the best baseline by __138%, 125%, 156%, and 11%__ for CORA, CITESEER, PUBMED, and OGB-COLLAB, respectively.

---

### Meta-Review · Area_Chair_i1wW · 2023-12-10

**Metareview:**

This paper proposes Gelato as a graph link prediction model, which addresses the issue of imbalanced class sizes that is very common in real-world graphs. The proposed method starts with node attributes learned from graph and uses a ranking loss to deal with class imbalances, as well as a negative sampling scheme to efficiently generate a small number of hard negative pairs.

The reviewers raised several concerns of this work, such as novelty, theoretic rigor, and hardness to tune the hyperparameters for such a complex model consisting of several parts.

**Justification For Why Not Higher Score:**

The issue of class imbalance is a classical problem, not specific to link prediction, therefore the proposed method does not look particularly novel.

The proposed method is mainly justified by only empirical results.

**Justification For Why Not Lower Score:**

N/A

---

### Decision · Program_Chairs · 2024-01-16

Reject